# Epidemiology of Turbot (*Scophthalmus maeoticus*) Bacterial Contamination, a Fishery Limiting Factor on the Romanian Black Sea

Aurelia Țoțoiu [1,†], Neculai Patriche [2], Victor Niță [1,†], Elena Sîrbu [2], Floricel Maricel Dima [2,3,*], Magda Ioana Nenciu [1] and Veta Nistor [2]

1 National Institute for Marine Research and Development "Grigore Antipa", 300 Mamaia Blvd., 900581 Constanța, Romania; atotoiu@alpha.rmri.ro (A.Ț.); vnita@alpha.rmri.ro (V.N.); mnenciu@alpha.rmri.ro (M.I.N.)
2 Institute for Research and Development in Aquatic Ecology, Fishing and Aquaculture, 54 Portului Street, 800211 Galati, Romania; patriche.neculai@asas-icdeapa.ro (N.P.); sirbu.elena@asas-icdeapa.ro (E.S.); nistor.veta@asas-icdeapa.ro (V.N.)
3 Faculty of Engineering and Agronomy in Braila, "Dunarea de Jos" University of Galati, Domnească Street, No. 111, 800008 Galați, Romania
* Correspondence: dima.floricel.maricel@asas-icdeapa.ro
† These authors contributed equally to this work.

**Abstract:** The aim of this research was to evaluate the health status of the *Scophthalmus maeoticus* (Pallas, 1814) population from the Romanian marine area in 2016–2019, by identifying and investigating information about bacterial and constitutional diseases, establishing the influence of these bacterioses on the researched populations, and highlighting the main biological disorders (reproduction, growth, and feeding) that manifested in the analyzed fish. The bacterial diseases reported in *S. maeoticus* (Pallas, 1814) populations were caused by pathogens from the genera *Vibrio*, *Aeromonas*, and *Pseudomonas*. Numerous skin diseases, especially various types of "wounds", lesions of bony tubercles, and tissue lesions may be the subsequent cause of neoplasms, as a result of fishing gear manipulations. The appearance of significant changes in the prevalence of neoplasia on the Romanian Black Sea coast can be considered an indicator of chronic stress (anthropogenic impact), rather than acute (environmental impact), and we suggest that the species could be used as a biological indicator of changes that may occur in the habitat in which it lives. In terms of future research directions, a combined analysis of the population structure, morphology and diseases determined in *S. maeoticus* populations, with a structural analysis of the habitat and bacteria contamination degree would be useful; this monitoring should be carried out regularly, to reveal changes in the Black Sea ecosystem, and to propose possible recommendations and protective measures.

**Keywords:** fish; disease; microorganisms; neoplasia; pathogen; impact

**Key Contribution:** The novelty of this work consists of approaching a unstudied niche on the Romanian Black Sea coast, by providing data regarding the degree of disease in *S. maeoticus*, and by creating, for the first time, synoptic ichthyo-pathological maps that could indicate the periods and areas of exploitation of this valuable aquatic resource.

## 1. Introduction

An important phenomenon in the marine ecosystem, which can provide a wealth of new information on the global level regarding the state of a population, is represented by disease. Diseases play an important role in the development of fish populations, especially commercial ones. The disease state represents a complex mix of phenomena and organic manifestations in the interrelationships with one or more pathogens, from the moment of contact with the host until the disappearance of the consequences. The presence of

disease in fish populations subjected to industrial fishing can represent a major and limiting factor, leading over time to a decrease in the viability of new generations, and producing changes at the population and ecosystem level. Diseases play an important role in the evolution of fish populations, especially commercial ones and, in recent decades, this can be demonstrated much more easily and precisely due to new technologies and determination methods. The continuous surveillance of diseases in fish in the natural environment, with special attention to flatfish, has been a new direction of research in recent decades, especially in the Baltic Sea, and recently in the Black Sea [1–3]. However, despite the fact that the Black Sea is overexploited, and subjected to intensive pollution, overfishing, oil extraction, and human activities, no studies on fish diseases are currently being carried out, except, sporadically, ones in connection with other research. Long-term research programs in the Black Sea area have mostly focused on parasitic diseases in fish populations [3].

The turbot is a demersal species that inhabits the continental plateau located both in Romania's area of competence and at the Black Sea regional level, and represents an important segment of the regional fishing potential, in terms of the commercial interest and demand in the local and international market. *S. maeoticus* is a benthic marine species, typical of soft bottoms; the juveniles are found near the shore, on the sandy bottom an as they grow, they retreat to greater depths. Adults are found in winter at depths of 60–70 m; in spring (March–April), they approach the shore up to 18–30 m, for reproduction [4].

*S. maeoticus* is an endemic flatfish, and one of the most valuable species for fishing, which can be considered a key species for monitoring and analyzing the processes carried out in the marine environment because, during its development, it covers practically all the habitats in the Black Sea [5].

Marine organisms live in environments relatively rich in bacteria and other microorganisms. Seawater can function as a medium for both the transport and growth of microorganisms; thus, marine organisms share an ecosystem with the microorganisms responsible for their diseases. Most bacteria that cause disease in marine fish are opportunistic pathogens that are present as part of the normal microflora of seawater. A few pathogens are dependent on a living host for their propagation, for example, *Renibacterium salmoninarum* and *Mycobacterium* spp. [6,7].

Environmental parameters can also directly or indirectly influence opportunistic bacteria. Environmental variables known to have such effects include the temperature, pH, osmotic resistance, oxygen levels, and iron availability [8]. The regulation of bacterial populations is a complex process that is not yet fully known about and understood. Less than 1% of the total number of bacteria in seawater may be active and able to grow on or in laboratory media, with the discrepancy between the total number of bacteria and the viable number being due to the existence of large numbers of dormant, inactive, or viable bacteria [8].

The epidermal mucus layer represents the primary biological interface between fish and the aquatic environment. The mucus coat can be a site of adhesion for bacteria, but it has also been said to function to prevent bacteria from firmly attaching to the skin [9,10].

Studies on the identification of infectious–contagious diseases in marine fish have generally been reduced to an informational level, with gaps due to limited access to biological material affected by various infectious diseases. The only existing data on *S. maeoticus* diseases highlight the outbreak of vibriosis in wild Azov *S. maeoticus* (*Scophthalmus maeoticus torosus* Rathkes) in 1986 [11]. Bacterial diseases in Black Sea *S. maeoticus* were studied only for the purpose of its commercialization in Turkey [12].

IBSS researchers from the Sevastopol region conducted the first attempt to assess the health status of the *S. maeoticus* spawning population from 2007 to 2010. The preliminary health status analysis showed that at least 20% of adult fish showed clear signs of various diseases [13].

The constitutional diseases reported in fish were described by specialists as "pathological processes triggered by morphological or body disorders" [14]. Ichthyopathologies

were highlighted through the identification, over time, of some functional disorders of the fish body, or through its anatomical changes.

If infectious and parasitic diseases can appear under the impact of biotic or abiotic factors, the causes of constitutional diseases are under the direct dependence of genes and the environment.

Specialists have signaled, over time, that the agents that lead to the appearance of these constitutional diseases in general can be of a genetic, hormonal, or environmental nature (the water physico-chemical parameters in which the respective population develops in the first stages of life), and that these diseases can be inherited [14].

This paper presents the bacterial diseases and neoplasia present in turbot from the Romanian coast during 2016–2019, with these diseases representing a limiting factor for the population's evolution.

## 2. Materials and Methods

The epidemiological analysis of the bacterial contamination of Romanian coast *S. maeoticus* populations was carried out in three stages: the sampling of fish, the identification of bacterial diseases, and the statistical analysis of the obtained results.

### 2.1. Fish Sampling

The epidemiological determination of the bacterial contamination in *S. maeoticus* was carried out through the analysis of 35 specimens. Only specimens that showed visible signs of illness were taken, and brought to the ichthyo-pathology laboratory. Fish were collected from scientific survey trawls carried out during research expeditions in the Romanian marine area within the National Fisheries Data Collection Program (Figure 1a).

The biological material was fished using demersal trawl and turbot gillnets, between 2016 and 2019.

The demersal trawl is a truncated cone-shaped fishing tool, equipped with its own arming system, towed to the substrate level with the help of a ship, by means of connecting elements (line, intermediate, bridle, etc.) [15].

The gillnet is a piece of fishing gear, consisting of a single net wall, with a vertical operating position generated by the reinforcing elements provided at the top (rafts) and bottom (leads), intended for use in catching, via hooking and entangling, fish species (Figure 1b) that move in the water mass or at the substrate level [15].

### 2.2. Determination of Bacterial Diseases

The macroscopic examination, which is indicative, was carried out to identify skin lesions, and the changes produced by them, on the hosts. It was performed using the naked eye, with a magnifying glass, observing the body surface, the eyes, and the gills. With the use of scissors, the fish were carefully sectioned on the abdomen, so as not to damage the internal organs. Each organ was then observed separately, to highlight possible internal haemorrhages, colour changes in the affected organs, necrotic areas, and other changes visible to the naked eye (Figure 2).

The healthy turbot specimens presented a specific colouration (in young specimens, the upper part was grey, the colour of sand, with white and black spots; in adults, the colour was brown without spots; the lower part in most of the fished specimens was white in colour and, rarely, some specimens presented blackish spots). There were no pathological changes or parasites in the eyes and gills. The integument of the fished turbot showed no changes created by the action of physical agents (aggression and trauma), without injuries on the body or neoplasms. Diseased specimens had skin lesions, pale gills, haemorrhagic spots on the surface of the body and gills, and haemorrhages in the peritoneum and internal organs (Figure 2).

The identification of the bacteria present in the *S. maeoticus* population from the Romanian Black Sea coast was performed through inoculation from skin lesions and

affected organs (liver, kidney, tissue) on the usual culture media, agar, and simple broth (for *Aeromonas* and *Pseudomonas*), and peptone water alkaline (for *Vibrio*) (Figure 2).

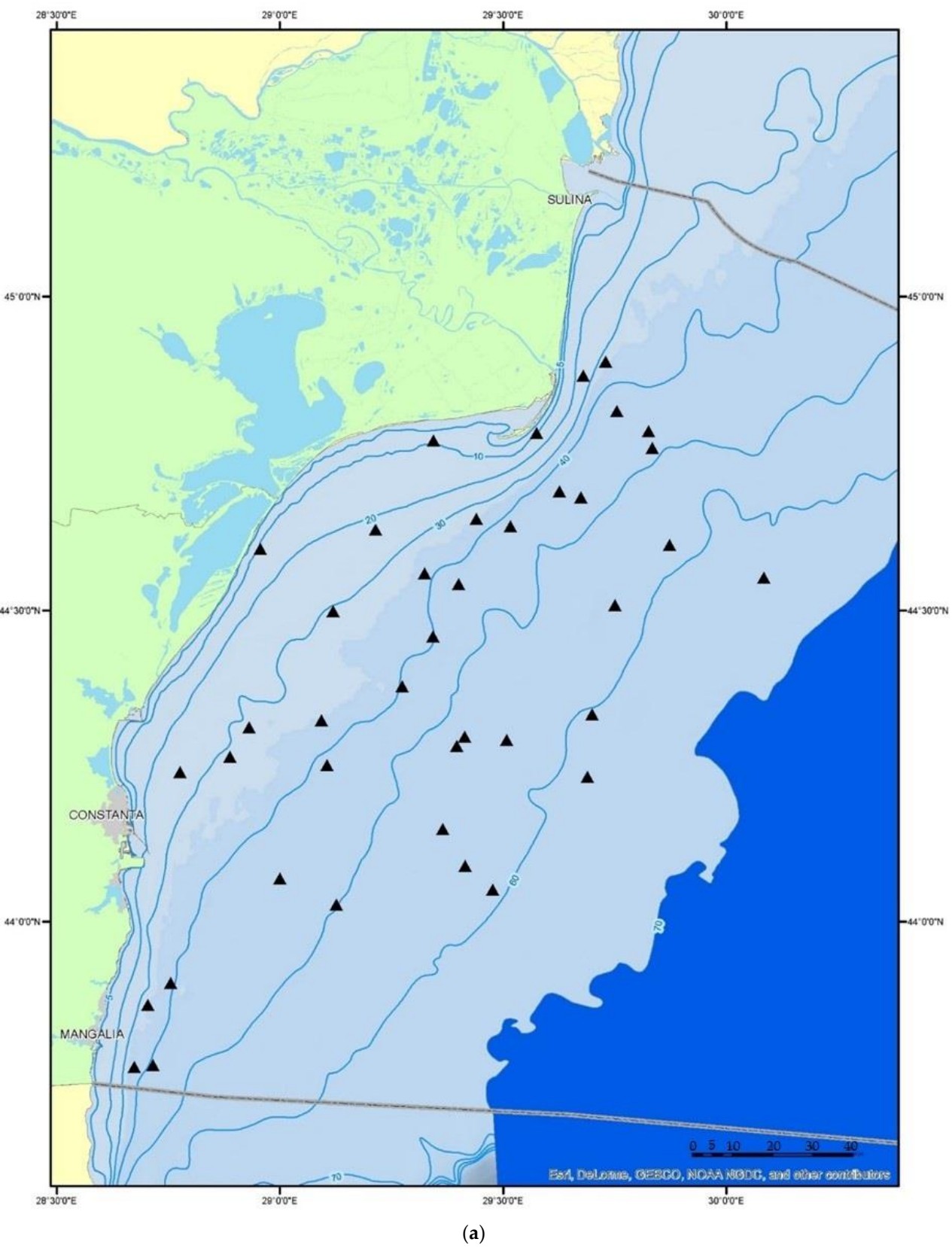

(**a**)

**Figure 1.** *Cont.*

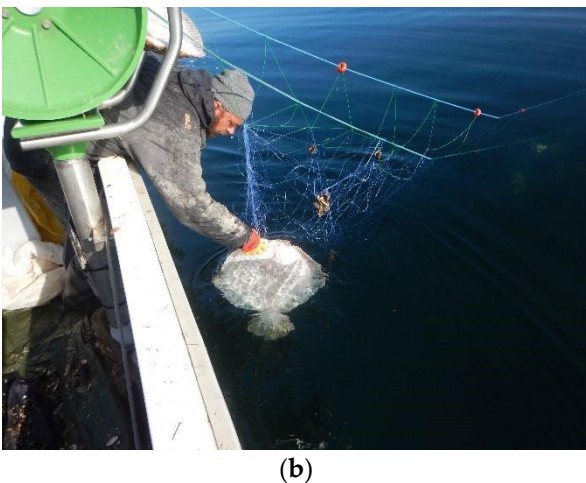

(**b**)

**Figure 1.** Distribution of trawling/sampling points for *S. maeoticus* during 2016–2019 (**a**); turbot gillnet (**b**).

From the initially inoculated cultures that developed on these media and later, replicates were made on the selective media for identification, according to Table 1.

From the specific colonies grown on the selective media, coloured smears were produced, using the Gram method, after which specific biochemical tests were performed for each group of bacteria. Depending on the characteristics of the colonies, the shape, and the size of the bacteria isolated, as well as the results of the tests, the bacteria involved in various fish infections were identified.

The protocol used for the determination of germ numbers is described in the paper "Bacteria from Fish and Other Aquatic Animals—A Practical Identification Manual" [16].

The Zeiss Axio Imagea A1 Microscope equipped with a camera (Jena, Germany) was used to determine the bacteria.

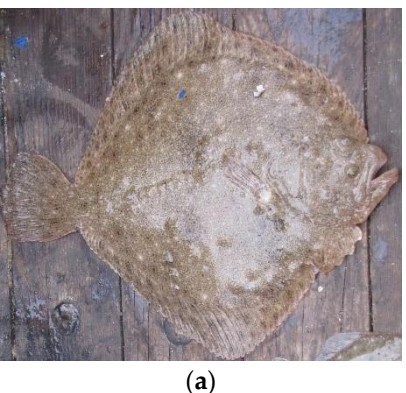 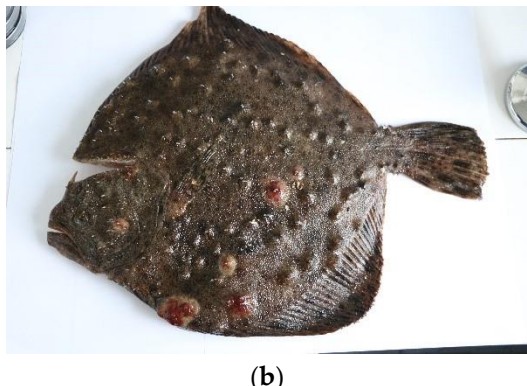

(**a**)               (**b**)

**Figure 2.** *Cont.*

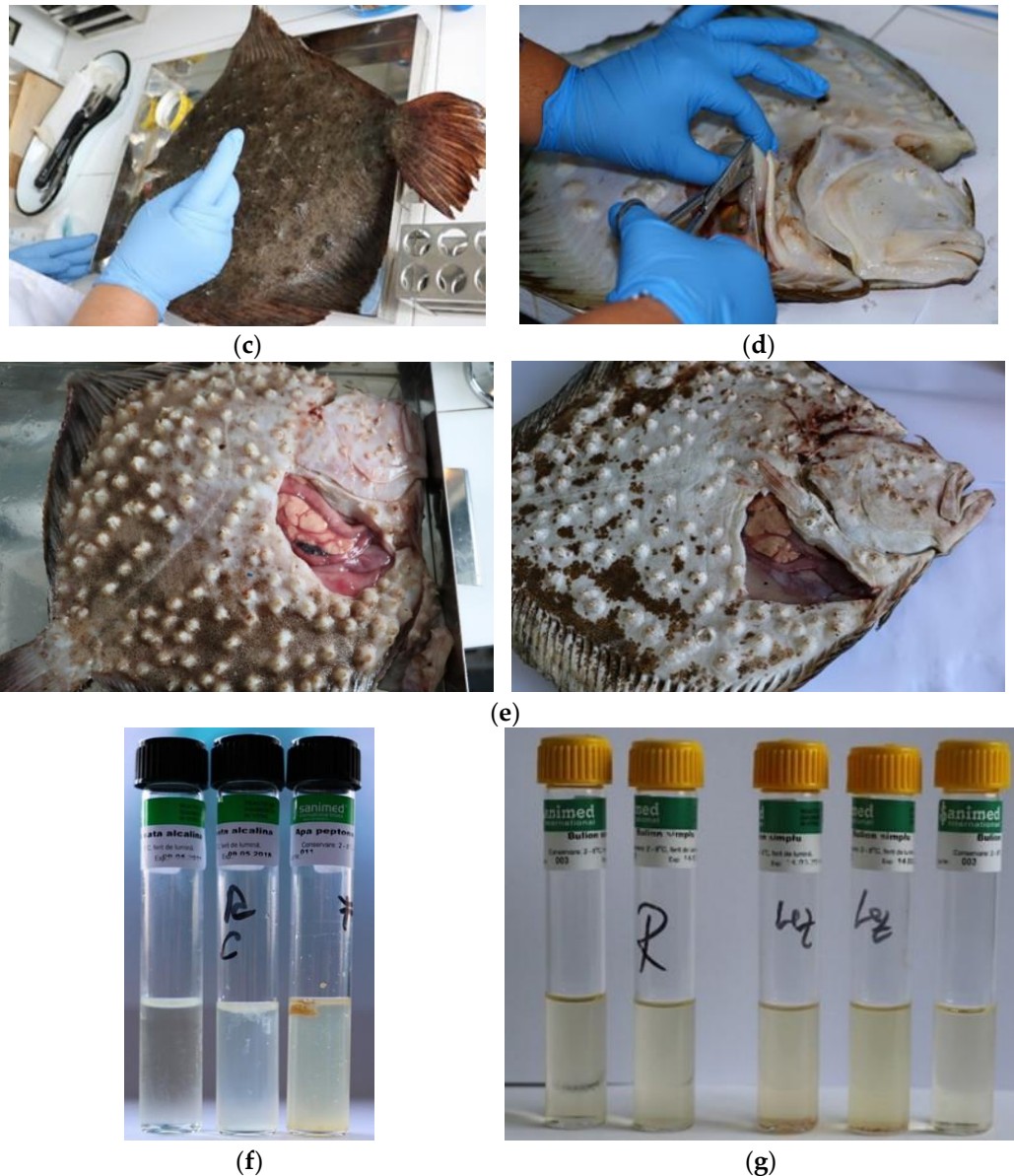

**Figure 2.** (**a**) healthy fish; (**b**) diseased fish; (**c**) inoculation from the lesion; (**d**) dissection of fish; (**e**) observation of internal organs; (**f**) the development of bacterial colonies in alkaline peptone water culture medium; (**g**) the development of bacterial colonies in simple broth culture medium.

Biochemical tests used to identify several characteristics of the developed colonies:

The oxidase test is tested directly from the culture; it is performed with the help of oxidase strips, which are moistened with distilled water, spread on a slide and, with the help of a Pasteur pipette, sealed at the tip. If the band turns blue in 15–30 s, the oxidase is positive (Figure 3). The strip is flamed, and the edge is placed directly into the disinfection mixture.

**Table 1.** Morpho-physiological characteristics of the main bacterial species of the genera *Vibrio*, *Aeromonas* and *Pseudomonas*, which cause infections in marine fish.

|  | *Vibrio anguillarum* | *Aeromonas hydrophila* | *Pseudomonas fluorescens* |
|---|---|---|---|
| **Colony characteristics** | Round colonies, 1–2 mm in diameter, transparent, smooth convex, shiny, with an opaque halo, soft consistency, grey–brown–yellow. | Round colonies, large with regular margins, smooth, opaque, white–grey to yellowish, up to 5 mm in diameter. | Small colonies, 2–3 mm diameter, with regular edges, slightly convex, dirty white, green, opaque, or slightly transparent |
| **Pigments in broth** | Stir the alkaline peptone water and the broth with the formation of a film on top. | Intense, uniform, uncharacteristic turbidity, without pigment or with brown pigment. Sedimentation tendency. | Turbidity, film on the surface, sometimes green, red, brown, orange pigment. |
| **Selective media** | TGBS, BSA—small colonies, round, transparent, bright, smooth, soft. | MK—red, large colonies. Colour changes due to the fermentation of sugars: red colonies on MacConkey, red-purple colonies on Levine, brown on Tryptic Soy Agar. | MI—blue-green colonies, with or without a black dot. It does not ferment the sugars, and does not cause discolouration on Levine. |
| **Morphological characteristics** | Bacilli slightly curved, 0.4–0.6/1–3 μm | Bacilli, coccobacilli, with polar flagellum. | Bacilli, coccobacilli (short, thick rods, with 0.5–0.6/1.5μm) |
| **Gram coloration** | - | - | - |
| **Mobility** | - | + | + |
| **Glucose** | +F | +F | +O |
| **Lactose** | - | + | - |
| **Sucrose** | + | + | +<br>- |
| **Indole** | + | + | - |
| **Lysine** | - | - | - |
| **Urea** | - | - | + |
| **H$_2$S** | - | + | - |

The culture media presented above, as well as those intended for tests, are commercially available ready-made. They were purchased from the company SANIMED INTERNATIONAL IMPEX S.R.L., Bucharest, Romania ("-" negative; "+" pozitive).

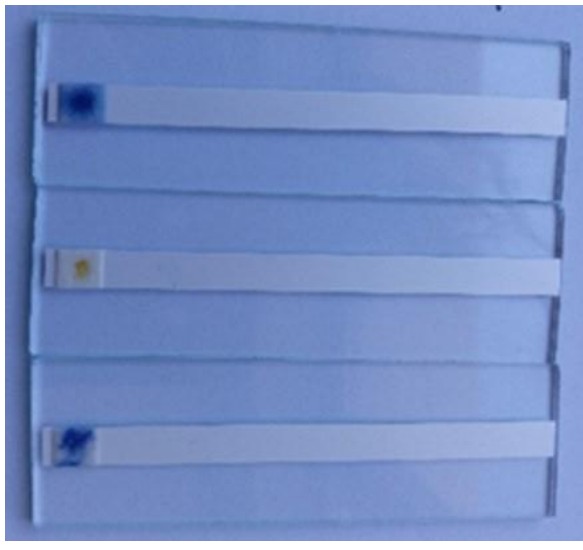

**Figure 3.** The oxidase test.

The mobility test, indole, H₂S, and urea, are conducted using the MIU medium. The medium is poured into small tubes, 1.5 mL each, into the column and, after solidification, it is pricked with the loop that was previously passed through the culture to be analyzed. On top of the medium, an indole strip fixed with the tube stopper is placed. If the indole band turns red, the reaction is positive; if it does not change colour, it is negative. Hydrogen sulphide is present when the puncture line turns black; if the culture spreads in the medium, the bacteria are mobile; and if the colour of the medium turns from yellow to red, the urea reaction is positive (Figure 4).

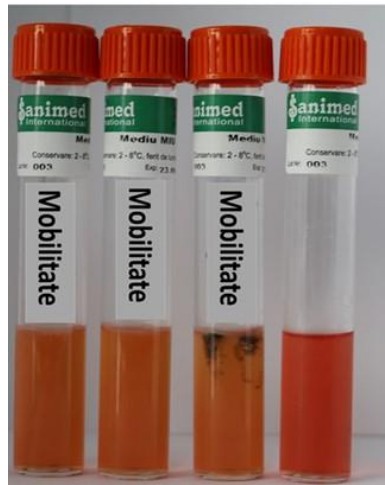

**Figure 4.** The mobility test.

The reaction to sugars is highlighted using semi-transparent pink triple sugar iron (TSI) medium. It is poured into inclined tubes and inoculated with the loop. If the colour changes to yellow for the inclined portion, it means that the bacteria have a metabolic action on glucose, lactose, and sucrose with gas production. If the black colour appears on the right portion, then the bacteria produce H₂S, and if the colour changes to yellow with or without gas, then the bacteria degrade glucose fermentatively (Figure 5).

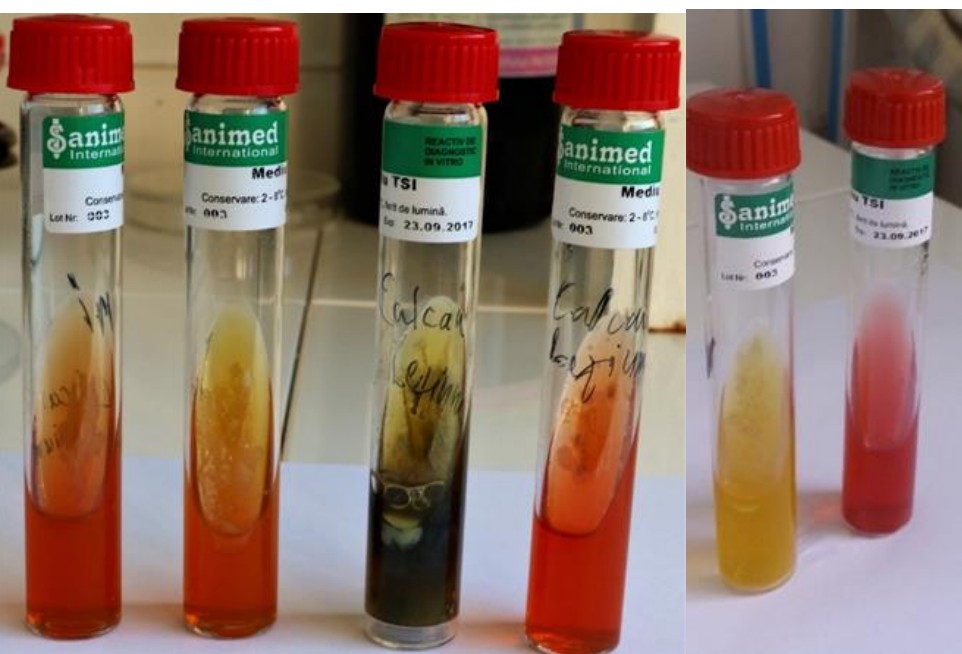

**Figure 5.** Triple sugar iron.

The main morpho-physiological characteristics of bacteria from the genera *Vibrio* (*V. anguillarum*), *Aeromonas* (*A. hydrophila*), and *Pseudomonas* (*P. fluorescens*), which are involved in marine fish infections, were identified and centralized from the microbiology treaties and ichthyo-pathology [17,18], as shown in Table 1.

Excel and PRIMER 7 software were used for data processing and statistical analysis, and the distribution maps were obtained using ArcGIS 10.6, (Tom Sawyer Software, Berkeley, CA, USA) software.

### 2.3. Determination of Constitutional Diseases

Constitutional diseases also manifest in marine fish, with the most important being malformations (which change the normal shape of the fish) and neoplasia (tumour processes, benign or malignant, affecting various organs and/or tissues).

Constitutional diseases are identified macroscopically, with malformations and tumour formations being large, visible to the naked eye and microscopically and, in the case of tumours, their characteristic structure being easily highlighted. Regarding neoplasia, the citations of various authors about the skin tumours affecting *S. maeoticus* from the Romanian coast are noteworthy [19].

In order to evaluate the presence of neoplasia in the turbot population, the prevalence was calculated—the percentage of infected fish from the total number of fish caught/year.

## 3. Results

### 3.1. Bacterial Diseases and Their Impact on S. maeoticus Populations from the Romanian Coast

Following the pathological determinations carried out in the laboratory, bacteriosis was identified in *S. maeoticus* populations in the natural environment, caused by pathogens from the genera *Vibrio*, *Aeromonas*, and *Pseudomonas*.

#### 3.1.1. Bacterial Contamination of S. maeoticus with V. anguillarium

*V. anguillarium* is the pathogen that causes the most-reported bacterial infection, namely vibriosis, with the disease being frequently described in marine fish from all over the world [20]. The etiological agent of the disease represented by the species *Vibrio anguillarum*, *Vibrio parahaemoliticus*, *Vibrio alginoliticus*, *Vibrio* sp., has been identified widely in over 45 species of fish, including important economic species. The isolation of bacteria of the genus *Vibrio* began with *Vibrio cholerae* in 1883, isolated by Robert Koch, which originates in humans and causes human infections, while aquatic *Vibrio* species, such as *Vibrio fischeri* and *Vibrio splendidus*, began to be cultivated in the late 1880s by Martinus Beijerinck [21].

*V. anguillarum* was first isolated by Canestrini in 1893, and was designated as *Bacterium anguillarum*, but later renamed as *V. anguillarum* in 1909 by Bergman [21]. *V. anguillarum* Bergmann, 1909 is a Gram-negative, halophilic straight or slightly curved rod-shaped, monotrichous, asporulate, encapsulated bacterium, measuring $0.5 \times 1.5$–$2.5/\mu m$. The length of the cilia is 2–3 times greater than that of the cell [17,18]. It is a eurythermal bacterium that can grow in a wide range of temperatures, from 10 to 35–42 °C, with the optimum growth temperature of 25–30 °C, and pH values of 6.9–9.6 [22]. Maeda et al., in 2003, associated the profiling of *Vibrio* bacteria in seawater with the seasonal increase in temperature [23]. Vibriosis, like most infectious diseases in ectothermic aquatic animals, usually occurs at a temperature lower than the optimal growth temperature of *V. anguillarium* [24].

Infection occurs through the penetration of the integument, although continuous renewal of the mucus layer prevents the adhesion of bacteria to epithelial cells. Existing lesions at the level of the integument, or the deterioration of the mucus layer are the main ports of entry for *V. anguillarum* [25,26].

Another way for bacteria to enter the host's body can be the oral route. In 1984, Larsen and Mellergaard highlighted that *V. anguillarum* is able to survive the acidic environment in the stomach [27], and Spanggaard et al. in 2000 noted that the gastrointestinal tract

of fish larvae is not fully developed, with a pH that is not low enough to inactivate the bacteria, and when an outbreak of vibriosis is reported, *V. anguillarum* is detected as the main pathogen [28].

In this study, the diagnosis was specified through the combination of the clinical examination with the bacteriological one, vibriosis being reported both in a chronic and acute form. The infection located at the integumentary level (chronic form) was reported in 29 of the 35 specimens analyzed. The acute form of vibriosis was characterized by skin lesions, pale gills, and infections localized in the intestine.

During the study period, the bacterial load with *V. anguillarum* varied from one specimen to another, depending on the collection period, depth, fish age, and natural immunity.

In 2016, the highest *V. anguillarum* number was recorded at stations 27, 28, and 33, with values between $4 \times 10^3$ CFU/g and $8 \times 10^3$ CFU/g in the autumn season, in fish collected from a depth of 64 m. We mention that the specimens fished in these areas were young specimens (4:4+–5:5+ years), with a lower natural immunity, and a high predisposition to illness if different forms of stress occur, determined by several factors: chemical—pollution, sea water quality; biological—competition for food; physical—temperature changes, etc.) (Figures 6 and 7).

Among *S. maeoticus* specimens collected and analyzed in 2017, a higher number of bacteria was reported than in the previous year, with the maximum bacterial load in fish from stations 2 ($4 \times 10^3$ CFU/g), 3 ($1 \times 10^4$ CFU/g) and 11 ($2 \times 10^4$ CFU/g), where the depth was between 38 and 61 m, and we add that the specimens analyzed were large, and showed visible signs of illness (Figure 6).

The microbial colonies of *V. anguillarum*, in 2018, recorded the highest values in the autumn season in *S. maeoticus* from stations 3 ($5 \times 10^3$ CFU/g), 12 ($7 \times 10^3$ CFU/g), 27 ($2 \times 10^4$ CFU/g), and 33 ($1 \times 10^4$ CFU/g). The fish age varied from one season to another; in the autumn season the youth specimens (5:5+ years) were predominant, presenting a higher risk of illness due to a lower immunity and to natural environment stress (Figures 6 and 7).

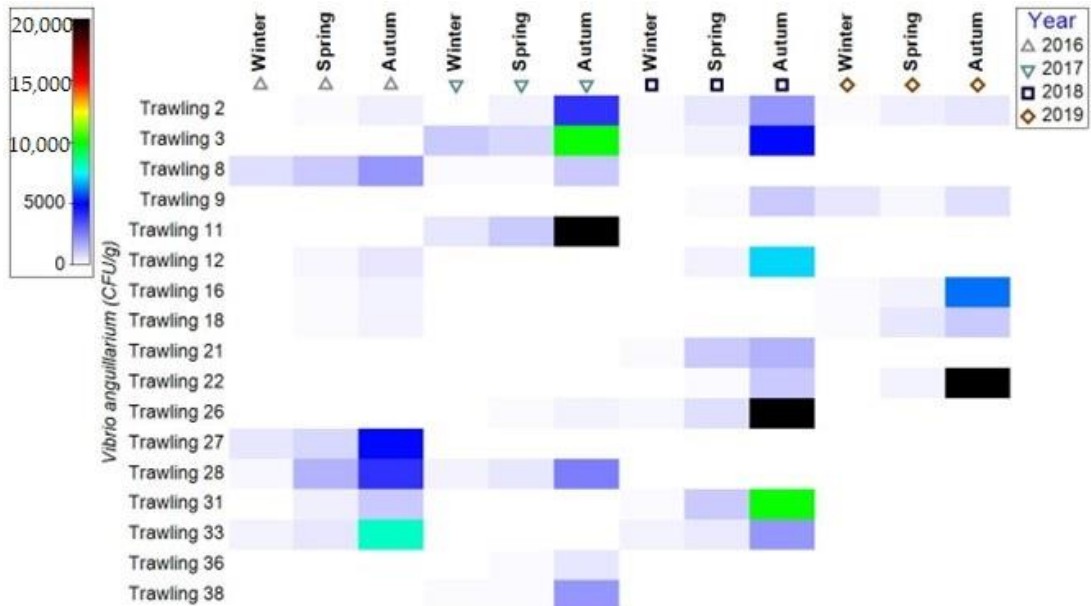

**Figure 6.** Shade plot with the number of *V. anguillarum* colonies (CFU/g) between 2016 and 2019 (number of colonies/specimens/trawling/season/year).

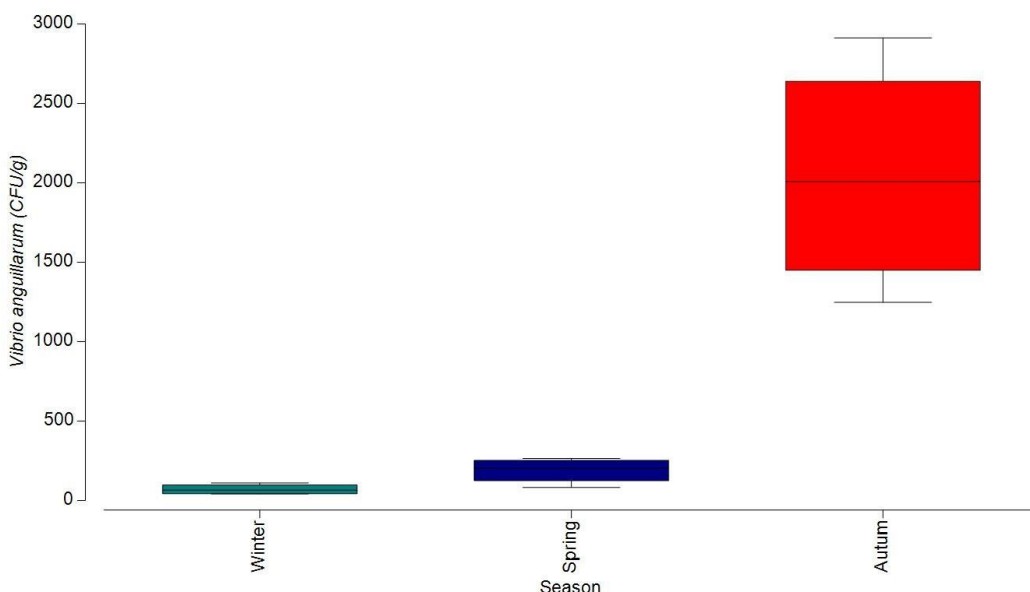

**Figure 7.** Box plot per season of the bacterium *V. anguillarum* (CFU/g mean value) during 2016–2019 (no statistically significant differences, *p* > 0.05).

In 2019, a lower number of *S. maeoticus* were analyzed, and the bacterial contamination reached high values in two stations (18–37 m depth and 22–58 m depth). The highest number of *V. anguillarum* bacteria was $2 \times 10^4$ CFU/g at station 22. *S. maeoticus* showing a chronic form of the disease, with multiple skin lesions, and infection at the intestinal level (Figure 6).

From a seasonal point of view, it was noted that the bacterial load with *V. anguillarum* had the highest number-of-colonies value in the autumn season, when the *S. maeoticus* population returned from depths of 50–90 m in the coastal waters for intensive feeding and preparation for wintering. The lowest values were reported in the summer season, when *S. maeoticus* retreats to greater depths and carries out limited feeding (Figure 7). These bacterial loads can be recorded in the natural environment, at the level of fish populations, and due to injuries caused by fishing gear. Fishing gear, through their mechanical action, can act on the whole body or parts of the fish's body, through pressure, piercing, and sectioning.

The effects of trauma can be multiple, and of various degrees of severity, with the most important being the destruction of the skin integrity, and the production of compressions, haemorrhages, and injuries to some tissues and organs. This trauma facilitates the penetration of pathogenic bacteria that adhere to the surface of the body or are present in water or sediment, constituting, together with the stress created, the starting point of some infections.

The number for the temporal distribution of *V. anguillarum* colonies varied from one year to another. The highest number was recorded in 2018; this year, large *S. maeoticus* specimens in a chronic state of the disease were identified. The fish were weakened, with multiple skin lesions, the internal organs were affected, and the digestive tube was completely deprived of food, and full of parasites (*Bothriocephalus scorpii*) (Figures 6 and 8).

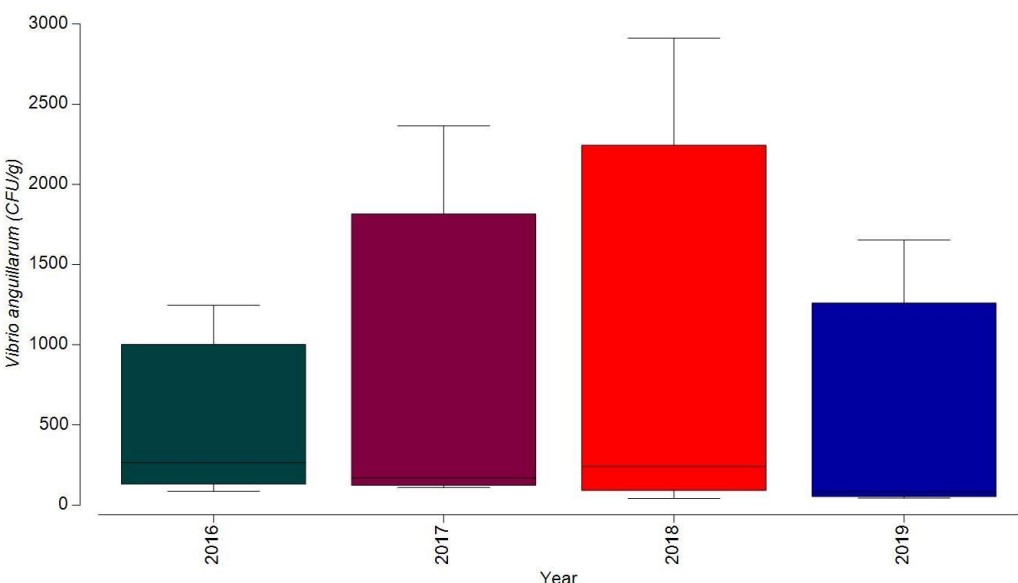

**Figure 8.** Box plot per year of *V. anguillarum* bacteria (CFU/g mean value) during 2016–2019 (no statistically significant differences, *p* > 0.05).

The bacterial load with *V. anguillarum* in 2017 was lower than in 2018, but higher than in the other two years studied. This year, a smaller number of specimens in a chronic condition were reported, and those in an acute condition showed a lower bacterial contamination. In 2016 and 2019, only specimens in an acute state of disease were reported (Figures 6 and 8).

The bacterial load with *V. anguillarum* in 2017 was lower than in 2018, but higher than in the other two years studied. This year, a smaller number of specimens in a chronic condition were reported, and those in an acute condition showed a lower virulence. In 2016 and 2019, only specimens in an acute state of the disease were reported, showing a lower virulence (Figures 6 and 8).

3.1.2. Bacterial Contamination of *S. maeoticus* with *A. hydrophilla*

According to Euzeby (1993), the ecological niche of aeromonads is aquatic fauna, especially fish and frogs, on whose mucosa they live as commensals and can cause infections. The increase in the number of aeromonads depends on the presence of organic matter [29]. They can cause a wide range of infections in wild and cultured fish species [30]. *A. hydrophila* is a mesophilic bacterium, which is part of the normal flora of the aquatic environment in which fish and other aquatic creatures live, but is also found in their gastrointestinal tract [31,32].

*A. hydrophila* is a pathogenic bacterium causing infections in *S. maeoticus* populations in the marine environment when environmental conditions deteriorate. The etiological agent of the disease is represented by species of the genus *Aeromonas—Aeromonas* spp., and *Aeromonas hydrophila* Zimmermann, 1890. *A. hydrophila* is a short rod-shaped, 0.3–1.0 × 1.0–3.5 μm, Gram-negative, motile, oxidase-negative, anaerobic, and facultatively anaerobic bacterium, producing gas from glucose, H2S positive. It can also show coccoid shapes, arranged in pairs, chains, or clusters [17,18].

Between 2016 and 2019, infection with *A. hydrophila* was reported frequently, but with less frequency than *V. anguillarum*. As in the case of *V. anguillarum*, the number of colony-forming units varied depending on the sampling period, depth, fish age, and acquired natural immunity. The infection was reported in 23 specimens out of the 35 specimens analyzed. The highest number of *A. hydrophila* colonies in 2016 was recorded at station 31 ($6.5 \times 10^3$ CFU/g), in the autumn season, and the lowest number of bacteria was noted at station 2 (25 CFU/g), in the spring season (Figures 9–11).

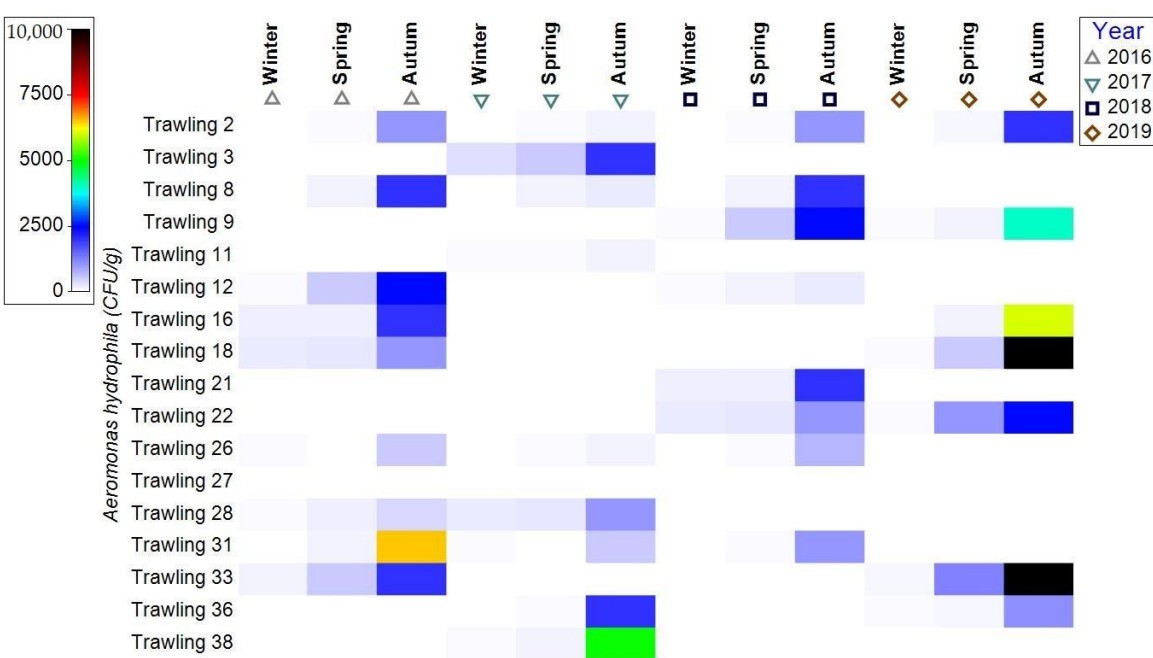

**Figure 9.** Shade plot with the number of *A. hydrophila* colonies (CFU/g) between 2016 and 2019 (number of colonies/specimens/trawling/season/year).

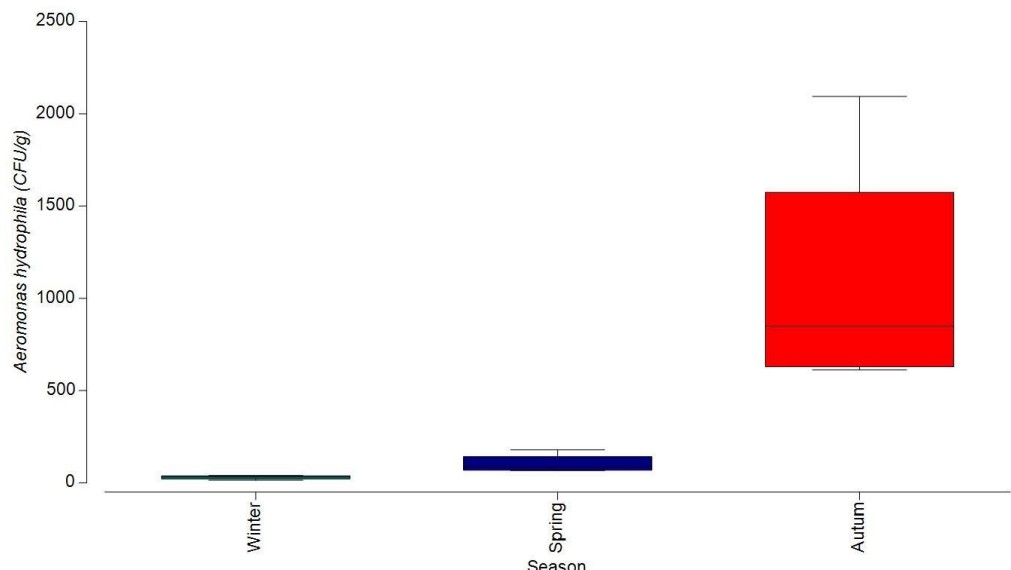

**Figure 10.** Box plot per season of the pathogen *A. hydrophila* (CFU/g mean value) during 2016–2019 (no statistically significant differences, $p > 0.05$).

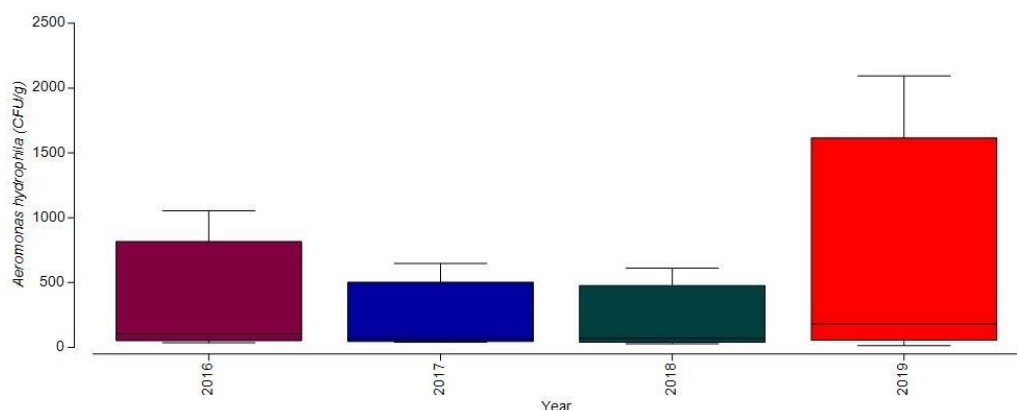

**Figure 11.** Box plot per year of the pathogen *A. hydrophila* (CFU/g mean value) during 2016 and 2019 (no statistically significant differences, $p > 0.05$).

In 2017, the infection produced by *A. hydrophila* was present in a chronic state of the disease at station 38, in the autumn season, with the total number of colonies reaching the value of $5 \times 10^3$ CFU/g when the depth was 63 m. This year, the collected specimens were aged between 5:5+–9:9+ years, and presented skin ulcers at the skin level, and haemorrhages at the level of the anus (Figures 9–11).

The bacterial load in 2018 with *A. hydrophila* was the lowest of the entire study period; the maximum value was $2.5 \times 10^3$ CFU/g, at station 9, at a depth of 52 m. In the winter and spring seasons, the recorded values were between 10 and 5102 CFU/g (Figures 9–11). All the individuals studied presented the chronic form of the disease; from the obtained results, it can be observed that, during the two seasons, the infection was signaled in the early phase.

In 2019, *S. maeoticus*-infected individuals with the chronic form of the disease were recorded, presenting skin ulcers, lesions on the fins and gills, and a pale liver. The highest value for *A. hydrophila* colonies was recorded at stations 18 (37 m depth) and 33 (63 m depth) of $10 \times 10^4$ CFU/g. Higher values were also reported at stations 9 (52 m depth), respectively, $4 \times 10^3$ CFU/g, and 16 (48 m depth), $6 \times 10^3$ CFU/g (Figures 9–11).

Regarding infection with *A. hydrophila* according to season, throughout the analyzed period, this showed a higher degree of occurrence in the autumn season; in the winter season, a very low degree of intensity was recorded (Figure 10). Symptoms of the disease were manifested in the form of skin ulcers, haemorrhages on the head, fins and anus, and gills, and a pale liver.

### 3.1.3. Bacterial Contamination of *S. maeoticus* with *P. fluorescens*

The genus *Pseudomonas* is one of the most diverse genera, and its taxonomy has undergone many changes in recent years. It includes over 260 species, and its classification is based on several characteristics imposed by the increased diversity of microorganisms, with the goal being identification and etiological diagnosis [33]. Pseudomonads are considered one of the most important fish pathogens, being responsible for ulcer-type diseases, including the ulcerative syndrome *Pseudomonas* septicemia [34,35].

Microorganisms of the genus *Pseudomonas* are Gram-negative, non-fermentative bacilli (unlike enterobacteria, which ferment glucose), aerobic, and non-sporulating, and can be involved in infections at various locations in immunodeficient organisms. It is a Gram-negative bacterium, bacillary in shape, 0.3–0.6 × 0.8–2μm, grows at 4 °C, and has 1–4 flagella. It produces fluorescein and not pyocyanin [17,18]. The infestation occurs through the digestive tract. The bacteria that are found in the seawater and in the digestive tract of the fish invade the body, after the appearance of the stress state generally caused by environmental condition variations to which the fish body is not adapted [36,37].

In 2016–2019, the etiological agent of the disease was represented by the bacterium *P. flurorescens*, which was reported in 19 specimens of *S. maeoticus* out of the 35 investigated.

The highest number of colonies of *P. fluorescens*, in 2016, was recorded at station 2 (38 m depth)—$1 \times 10^3$ CFU/g, in the autumn season, and the lowest value was reported at station 33 (10 CFU/g) (Figures 12–14).

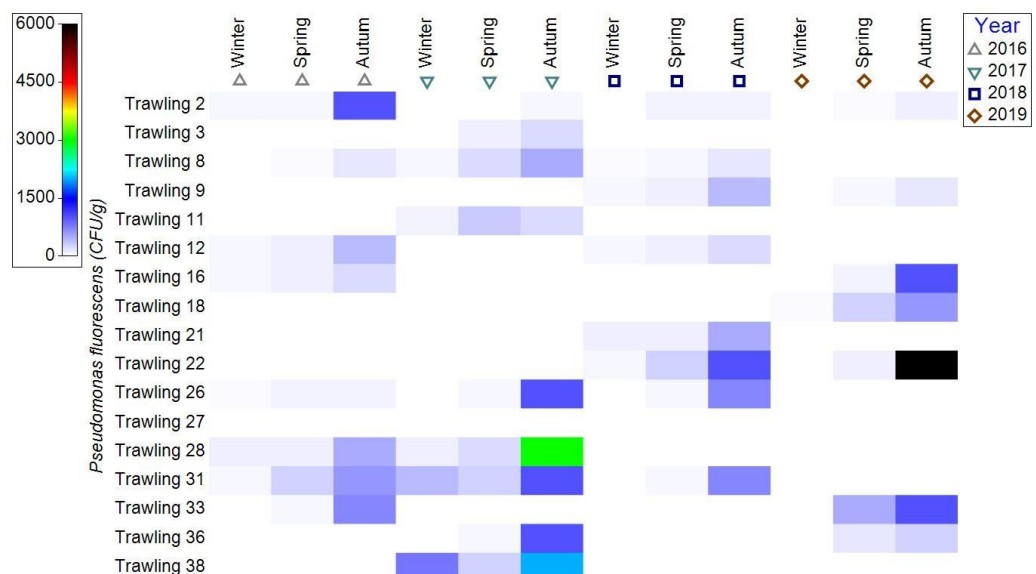

**Figure 12.** Shade plot with the number of *P. fluorescens* colonies (CFU/g) in 2016–2019 (number of colonies/specimens/trawling/season/year).

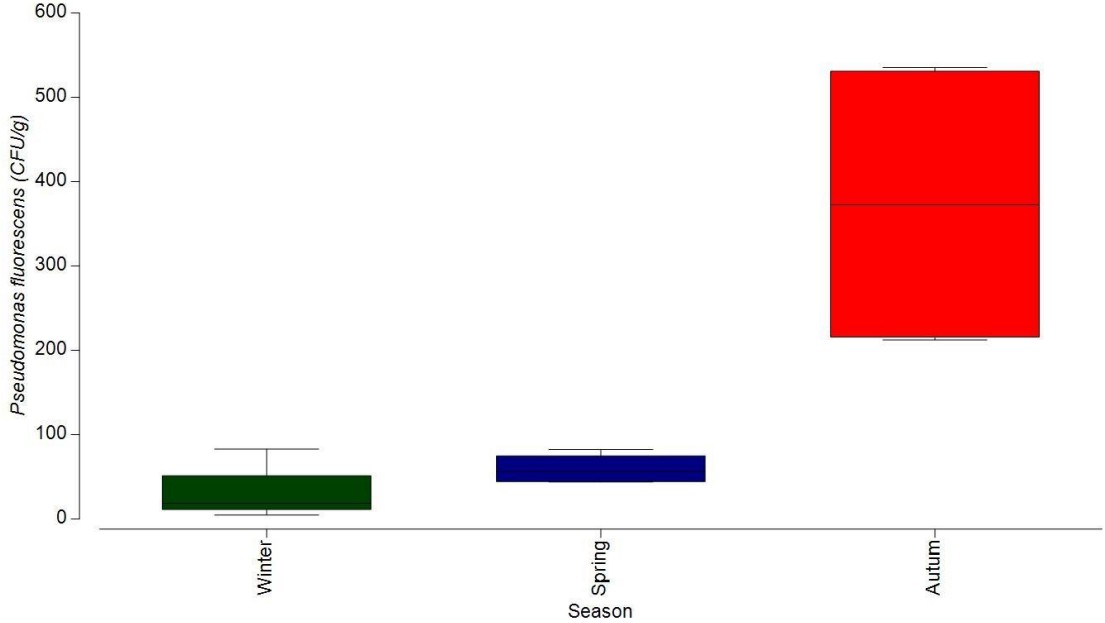

**Figure 13.** Box plot per season of the pathogen *P. fluorescens* (CFU/g mean value) during 2016–2019 (no statistically significant differences, $p > 0.05$).

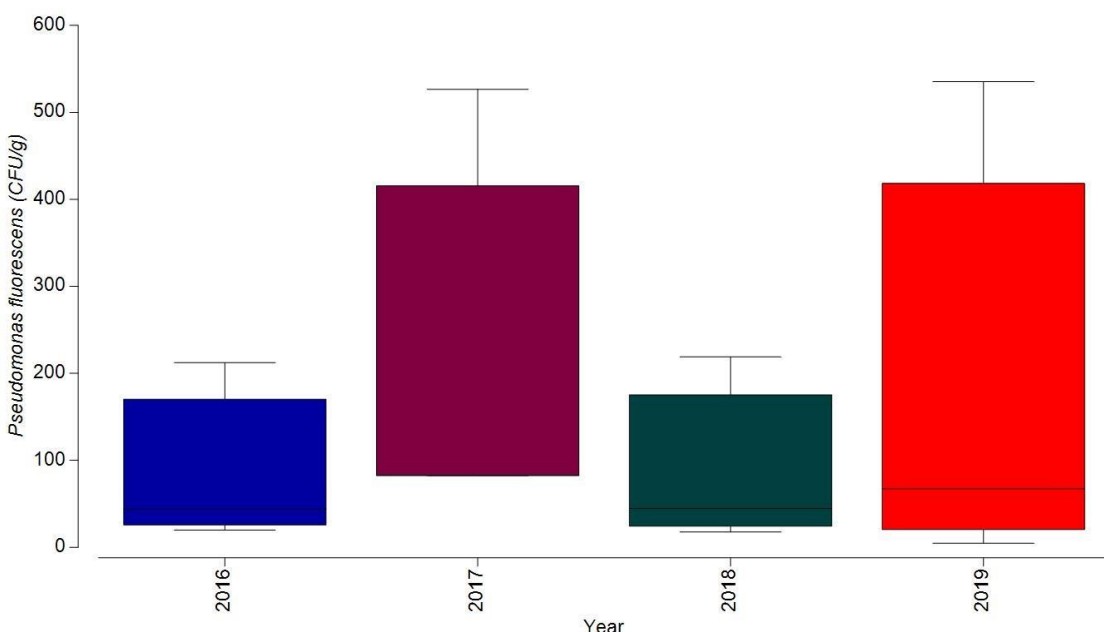

**Figure 14.** Box plot per year of the pathogen *P. fluorescens* (CFU/g mean value) during 2016 and 2019 (no statistically significant differences, *p* > 0.05).

The *S. maeoticus* analyzed in 2017 were the most infected with *P. fluorescens* bacteria. The samples analyzed in the autumn season showed the highest number of bacteria ($1 \times 10^3$–$3 \times 10^3$ CFU/g), and the fish showed signs of disease, and showed feeding disorders and a low degree of fattening (Figures 12–14). In the natural environment, the disease occurs through the digestive tract, and the first signs are manifested under the conditions of a state of stress. Affected specimens showed bleeding at the base of the fins, and on the body surface and palatal mucosa, and pale gills. As the state of the disease cannot be monitored in the natural environment, nor the impact on the fish stocks affected by this bacteriosis, we cannot know its evolution and extent in the investigated fish populations [38,39].

*P. fluorescens* reached its maximum value at station 22 ($1 \times 10^3$ CFU/g) in 2018; the *S. maeoticus* specimens investigated this year presented a low degree of bacterial contamination (Figures 12–14).

In 2019, the highest degree of infection with *P. fluorescens* of the entire study period was reported, with the bacterial contamination reaching the value of $6 \times 10^3$ CFU/g at station 22, at a depth of 58 m. The *S. maeoticus* specimens were also contaminated by *V. anguillarium* and *A. hydrophilla*, and the analyzed fish presented a chronic form of the disease, and a weakened organism, with visible feeding disorders (Figures 12–14).

From a seasonal point of view, *P. fluorescens* showed very low values in the winter–spring seasons, and in the autumn season it showed a much lower degree of infection than the other two identified bacterial species. The amount of organic matter and anthropogenic impact can generate this increased bacterial contamination rate in *S. maeoticus* [34,35].

The graphic representation of the *P. fluorescens* level of infection by year highlights that the highest values were recorded in 2019 and 2017. For the years 2016 and 2018, the average values recorded were about $2 \times 10^2$ CFU/g (Figure 14).

### 3.2. Constitutional Diseases—Evaluation of Neoplasia from the Romanian Coast in 2016–2019

Whereas infectious and parasitic diseases can appear due to many causes, of a biotic or abiotic nature, the causes of constitutional diseases directly depend on genes and the environment.

Malformations (deviations from the normal form of the fish) and neoplasia (tumour processes, benign or malignant, affecting various organs and/or tissues) are the most significant constitutional diseases reported in industrially fished fish.

Fishing gear has an impact on the health of turbot populations, generating immediate or long-term consequences. The pathological conditions caused in turbot populations by these physical agents during industrial fishing can include aggression and traumatization (traumas), which constitute sources of stress and diseases that can change the health status of the fish. Under the influence of these stressful environmental conditions, the resistance of the fish organism decreases, thus favoring the onset of constitutional diseases, and the appearance of neoplasia [40].

Neoplasms are represented by any increase in the volume of an organ or tissue generated by a continuous proliferation of cells, with a certain characteristic morphological change.

Robert R.J. defined neoplasia as a "new growth" formed through aberrant cell multiplication, resulting in an excessive number of cells [14].

The neoplasms in homeothermic animals are functionally and structurally identical to the tumours identified in fish. Tumours have a complex etiology, but many factors that lead to their appearance and growth are not known. However, there are also known factors that are involved in the formation of neoplasms in fish, namely chemical and biological toxins, viruses, different physical agents, hormones, and age, sex, genetic predisposition, and host immunity.

The tumour formation frequency is related to the geographical area, the living habitat of the species, and the quality of biotic and abiotic factors, but an important role is also played by the nature of the provocative agent.

Tumours (neoplasms) are among the constitutional diseases, and were reported in *S. maeoticus* populations from the Romanian Black Sea coast throughout the analyzed period (2016–2019). The numerous skin diseases appear primarily due to the appearance of various wounds at the integumentary level, bone button injuries, and tissue injuries. In general, these can also be the result of contact with fishing gear.

Investigations carried out during the tumour monitoring period revealed the presence of various signs of skin damage and inflammation present on both sides of the body, at an acute and chronic level. The reported inflammations ranged from minor haemorrhagic ulcerations in the skin, to its total destruction. In acute cases, the appearance of secondary infections added to the inflamed skin areas (Figure 15).

In the evolution of the tumour forms reported in the *S. maeoticus* population from the Romanian coast, an important role was probably played by the environmental conditions in the living habitat of the diseased specimens. If the living environment is contaminated, added to the reduction in the immune status of the fish, then the lesions can become chronic, which involves muscle and even bone tissue; later, these processes lead to tissue necrosis. The most frequently observed clinical signs during the study period in the diagnosed fish were erythema, haemorrhages, and ulcers. The disease begins with the appearance of a formation on the skin. Microorganisms were present only in the ulcerative tissues.

In specimens industrially fished in the Romanian marine area, the investigated tumours presented in various forms of evolution, namely:

- the early phase, when the epithelial surface of the center of the lesions is removed, and superficial hemorrhagic ulcers are produced;
- the phase with three distinct zones: a peripheral pale zone; a dark-grey intermediate zone; and a central hemorrhagic zone.

The prevalence of *S. maeoticus* specimens affected by neoplasia during the study period was 18.8%; the most affected specimens were reported in 2019, and the least in 2017 (10.4%) (Figure 16).

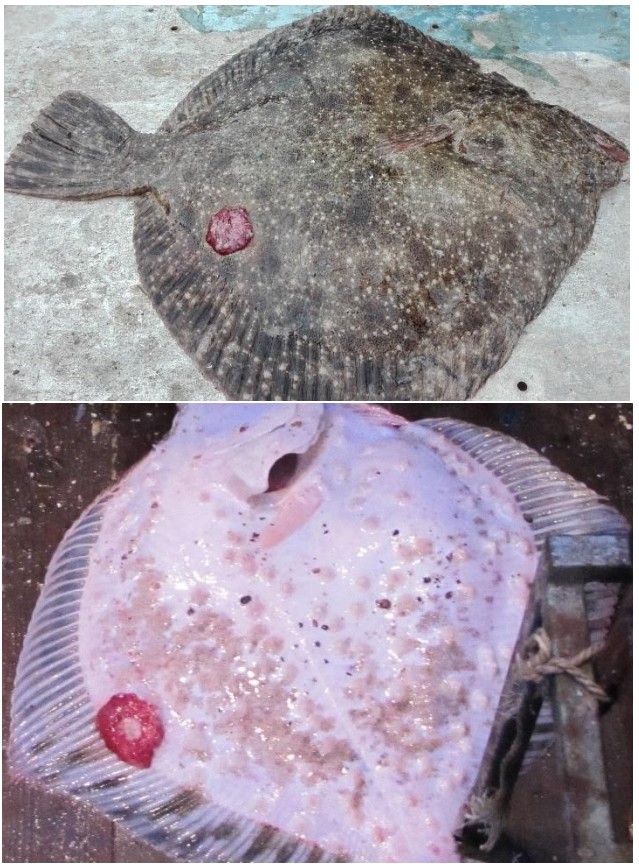

**Figure 15.** Tumours identified in the *S. maeoticus* population from the Romanian coast.

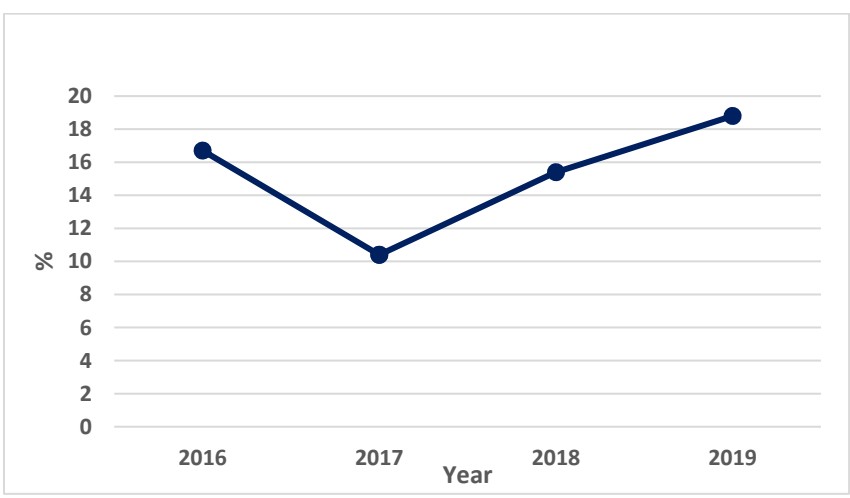

**Figure 16.** Prevalence of neoplasia in the *S. maeoticus* population, between 2016 and 2019.

Injuries such as "wounds" can be mechanical, caused by bottom-fishing gear combined with pollution and habitat degradation, with subsequent bacterial invasions, so it can be argued that this is a factor behind the high prevalence in the *S. maeoticus* population under investigation. Neoplasia can greatly affect the *S. maeoticus* population in this area, and individuals affected by this constitutional disease cannot be provided for human consumption.

## 4. Discussion

Over the decades, research on the Black Sea turbot population has mainly focused on biomass assessment in industrially fished stocks, with demersal trawls in different areas, calculated using mathematical models [41]. The health state has been evaluated less, and only in certain areas of the Black Sea.

The signs of disease within a population are visible mainly after the installation of the state of stress, which reduces the resistance of organisms, and favors the outbreak of infectious, parasitic, or other diseases [42]. The manifestation of the disease phenomenon can cause growth, reproductive, and nutrition disorders at the population level and, in many cases, these epidemiologies can cause significant mortality in fish populations [43].

Studies in the field of fish population pathology have experienced an increase in frequency and intensity in recent decades. Currently, disease research reports trophic relationships, but also their causes and methods of transmission in the food chain, as well as in humans [44,45]. The results of these studies have filled in knowledge gaps regarding the prevalence and distribution of diseases identified in fish populations.

Fish bacterioses in the natural environment are numerous, and some of them produce dramatic effects, such as epizootics or even mass mortality. The severity of a bacteriosis is directly related to the infected dose and the fish receptivity. Reported bacterial diseases can affect all age classes within a fish population of increased commercial value, with the transmission and severity of infectious–contagious diseases in aquatic ecosystems being accentuated in inappropriate environmental, feeding, stress, or physiological conditions. The natural and acquired resistance of fish populations in the natural environment are of particular importance to their evolution, with the diseases that appear being a consequence of the interaction of genetic and ecological factors [46].

Studies on the identification of infectious–contagious diseases in marine fish have generally been reduced to an informational level, with gaps due to the limited access to biological material affected by various infectious diseases. The only existing data on *S. maeoticus* diseases highlight the outbreak of vibriosis in wild Azov *S. maeoticus* (*Scophthalmus maeoticus torosus* Rathkes) in 1986 [11]. Bacterial diseases in Black Sea *S. maeoticus* were studied only for the purpose of its commercialization in Turkey [12].

The only updated study on fish diseases in the Black Sea was conducted along the coastline off Turkey between 2007 and 2009, to show the presence of the viral haemorrhagic septicemia virus in marine fish [47].

Researchers from the Biological Station of Sevastopol (BBS) conducted a first attempt to assess the health status of the *S. maeoticus* spawning population from 2007 to 2010. The preliminary health status analysis showed that at least 20% of adult fish showed clear signs of various diseases [3].

The detected pathogens, from the genera *Vibrio, Aeromonas* and *Pseudomonas*, are agents with a high infection rate, a fact that can be explained by the increase in environmental temperatures, but also by the amount of organic substances. This fact represents an alarm signal for the future, as research is needed in this direction.

These bacteria are widespread in marine water, but their pathogenic action is triggered when fish are weakened by the above-mentioned factors. Bacterioses affected, to a certain extent, up to 50% of the turbot specimens investigated during 2014–2017 [38]. Fish stocks can be affected by, and can reduce their number due to, these diseases, with the risk that they may spread to other populations in the habitat. Diseased fish show a low growth rate, nutritional disorders due to a lack of food in the stomach, and poorly developed gonads [42].

Recent studies have demonstrated that environmental stress could affect the expression of virulent genes that are involved in the ability of the bacterium *V. anguillarum* to cause infections [48,49]. From a spatial point of view, the bacterial load with *V. anguillarum* was reported both in the southern and the northern part of the Romanian marine area. The most affected *S. maeoticus* populations were those from the northern part throughout the study period; in this area, where those in a chronic state of the disease (vibriosis) were fished,

the environmental conditions and the anthropogenic impact helped these manifestations (e.g., pollution, excessive fishing) (Figures S1–S4 in Supplementary Material). Vibriosis, in acute epizootics, spreads rapidly, with most infected fish dying without showing clinical signs [41].

*A. hydrophila* is considered an opportunistic pathogen associated with haemorrhagic septicemia in cold-blooded animals, including amphibians, reptiles, fish, and seafood animals [30]. The pathogenicity factors include aerolysin, a pore-forming toxin, and a heat-stable, cytotoxic enterotoxin. It induces peptosis in infected macrophages [50].

The spatial distribution of the bacterial contamination with *A. hydrophila* in the turbot population from the Romanian coast throughout the study period recorded a high bacterial load in the southern part, as well as in the northern part, of the Romanian marine area. The most-affected turbot populations were those in the southern part; in this fishing area, especially in 2019, young specimens were in a chronic state of disease. Between-aged turbots predominated in the scientific fishing carried out in this marine area (Figures S4–S7 in Supplementary Material).

Like *A. hydrophila*, *P. fluorescens* is considered an opportunistic bacterium that causes septicemia in fish under stress or affected by other pathogens, in many cases acting as a primary fish pathogen. Between 2016 and 2019, the degree of infectious intensity with *P. fluorescens* recorded its highest values in the population from the northern part of the Romanian coast. The severity of bacterial diseases is very difficult to quantify in the natural environment; monitoring them further, and trying to reduce pollution and over-fishing in the affected areas, would be a recommendation in this case (Figures S8–S11 in Supplementary Material).

Fish populations, especially *S. maeoticus*, from the Romanian Black Sea coast, are affected by numerous constitutional diseases. In 1993, Dumitrescu E. found the presence of neoplasia in 20% of the *S. maeoticus* specimens studied, and pointed out that the presence of tumours can be a danger for this species [19]. During adult life, it has continuous contact with the sea bottom, and this fact makes the species more vulnerable to ulcerative skin lesions, with these diseases being recorded in different areas of the Black Sea [51].

Between 2007 and 2012, in the Sevastopol region, specialists monitored the epizootological status of Black Sea *S. maeoticus* diseases. A variety of constitutional diseases present in the *S. maeoticus* population in the area has been reported, namely: the inflammation of different parts of the integument, ulcerations, extensive necrosis of the epidermis, fin erosion, and various types of epithelial neoplasia [1].

Scientific research in the field of industrial marine fisheries has seen an increase in frequency and intensity in recent decades. The results of this research have led to the accumulation of new information regarding the factors affecting the abundance and distribution of fish populations. An environmental factor in the marine ecosystem, which can lead to a wealth of new information, and is of major importance, is represented by disease. Diseases play an important role in the evolution of fish populations, especially commercial ones. This can be demonstrated much more easily and precisely, thanks to new technologies and methods of determination in recent decades. Of great relevance is the research on diseases in trophic relationships, but also on their causes and ways of transmission, in the food chain, as well as in humans [52].

## 5. Conclusions

The high level of diseased fish may be related to the stress of environmental factors combined with various types of pollution, including chemical and organic. Of the three bacterial infections reported, the most frequent was vibriosis, the infection produced by *A. hydrophila* was reported with a lower infectious intensity, and the infection produced by *P. fluorescens* was the least frequent, and showed a minimal intensity in the investigated *S. maeoticus* specimens. In the natural environment, the state of the disease cannot be monitored, and nor can the impact on the fish stocks affected by these bacterioses, and we cannot fully know about their evolution and extent in the investigated wild fish pop-

ulations. However, these diseases can be a limiting factor causing feeding, growth, and reproductive disorders.

The appearance of significant changes in the prevalence of neoplasia on the Romanian coast can be considered an indicator of chronic stress (anthropogenic impact), rather than acute (environmental impact), and we propose that the species could be used as a biological indicator of the changes that may occur in the habitat in which it lives.

The novelty of this work consists of approaching an unstudied niche on the Romanian Black Sea coast, by providing data regarding the degree of disease in *S. maeoticus*, and by creating, for the first time, synoptic ichthyo-pathological maps that could indicate the periods and areas of exploitation of this valuable aquatic resource.

Future research directions include a combined analysis of the population structure, morphology, and diseases determined in *S. maeoticus* populations, with a structural analysis of the habitat and degree of contamination. This monitoring should be carried out regularly, to reveal changes in the Black Sea ecosystem, and to propose possible recommendations and protective measures.

**Supplementary Materials:** The following supporting information can be downloaded at: https://www.mdpi.com/article/10.3390/fishes8080418/s1, Supplementary Material: Ichthyo-pathological maps of bacterial infestation in Romanian Black Sea turbot. Figure S1: Number of *V. anguillarum* colonies (CFU/g) in 2016; Figure S2: Number of *V. anguillarum* colonies (CFU/g) in 2017; Figure S3: Number of *V. anguillarum* colonies (CFU/g) in 2018; Figure S4: Number of *V. anguillarum* colonies (CFU/g) in 2019; Figure S5: Number of *A. hydrophila* colonies (CFU/g) in 2016; Figure S6: Number of *A. hydrophila* colonies (CFU/g) in 2017; Figure S7: Number of *A. hydrophila* colonies (CFU/g) in 2018; Figure S8: Number of *A. hydrophila* colonies (CFU/g) in 2019; Figure S9: Number of *P. fluorescens* colonies (CFU/g) in 2016; Figure S10: Number of *P. fluorescens* colonies (CFU/g) in 2017; Figure S11. Number of *P. fluorescens* colonies (CFU/g) in 2018. Figure S12. Number of *P. fluorescens* colonies (CFU/g) in 2019.

**Author Contributions:** Conceptualization, A.Ț. and V.N. (Victor Niță); methodology, A.Ț. and E.S.; software, A.Ț. and E.S.; validation, V.N. (Victor Niță), F.M.D. and N.P.; investigation, A.Ț.; resources, A.Ț.; writing—original draft preparation, A.Ț.; writing—review and editing, V.N. (Victor Niță), M.I.N., N.P. and V.N. (Veta Nistor); visualization, V.N. (Victor Niță), M.I.N. and F.M.D.; supervision, N.P. All authors have read and agreed to the published version of the manuscript.

**Funding:** This research received no external funding.

**Institutional Review Board Statement:** The study was conducted in accordance with the guidelines of the EC Directive 86/609/EEC regarding the protection of animals used for experimental and other scientific purposes, and Romanian legislation (Law 43 of 11 April 2014), and was approved by the Ethical Commission of Academy of Agricultural and Forestry Sciences "Gheorghe Ionescu Sișești", Department of Veterinary Medicine (Decision No 7 on 9 February 2021).

**Data Availability Statement:** All the data are available from the first author, and can be delivered if required.

**Acknowledgments:** This research was performed within the Romanian National Fisheries Data Collection Program.

**Conflicts of Interest:** The authors declare that they have no known competing financial interest or personal relationship that might appear to influence the activity reported in this paper.

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
