# Peer review of "Epidemiology of Turbot (Scophthalmus maeoticus) Bacterial Contamination, a Fishery Limiting Factor on the Romanian Black Sea"

_fishes, doi:10.3390/fishes8080418_

Round 1
Reviewer 1 Report
In this study, authors describe " Epidemiology of turbot (Scophthalmus maeoticus) bacterial contamination, a fishery limiting factor at the Romanian Black Sea ". Based on this field, the Ms could be revised before being published in this magazine.
1. In Fig. 2, compare the healthy and the diseased fish, and describe them.
2. Add the label in each figure of Fig. 2, and describe them in the Ms.
3. Detail describ the Figures. For example. In Fig. 2,
4. An evolutionary tree of bacteria obtained should be added to evaluate the epidemiology of S. maeoticus
5. The results of physiological and biochemical experiments should transfer to the part of results.
6. Add the statistical analysis (e.g., significantly differences) in Fig. 7, 8,10,11,13, 14, and describe them in the Ms.
7. Line 389-391, cite the relationship between tumors and artificial fishing in the text
English is OK.
Author Response
Thank you for your review and the comments and recommendations you provided, which we have made and translated into the manuscript fishes-2532507_revised.

Reviewer 2 Report
This manuscript reports the evaluation of health status of a Scophthalmus maeoticus population from the Romanian Black Sea area in the period 2016-2019. During a three year period, the number of fish studied ought to be higher. It will be difficult to give safe conclusions based on the number of fish studied.
Line 40: evolution; I think that the authors could use development in places where evolution is used.
L 77: grammatical error
L 115: analyzing 35 specimens; The number should have been higher through a three year period.
L 210: S. maeoticus in italics. Please put all latin names in italics.
L 225: In 1994, Larsen; change to: In 1984, Larsen and Mellergaard (ref 26 L 640+641
L 238: virulence values; What do the authors mean by virulence values. Please check the definition of virulence!!
L 245: higher virulence; do the authors mead higher number of colonies??
L 275: Grammatical mistake (should be were (plural)
L 310: lowest virulence; change to: lowest bacterial load??
L 366: the virulence reaching ….6x103CFU/g; ??????
L 479: BSS researchers; abbreviations should be written in full the first time they are mentioned.
L 530+531: A variety….have been, grammatical error (singular)
English fine, just some grammatical errors.
Author Response
Thank you for your review and the comments and recommendations you provided, which we have made and translated into the manuscript fishes - 2532507_revised.

Round 2
Reviewer 2 Report
This manuscript reports the evaluation of health status of a Scophthalmus maeoticus population from the Romanian Black Sea area in the period 2016-2019. During a three year period, the number of fish studied ought to be higher. It will be difficult to give safe conclusions based on the number of fish studied. This fact is probably why no significant results were found. The observations made are valuable as such, but it is scientifically not so valuable due to the low number of observations (fish).